# R4D: Utilizing Reference Objects for Long-Range Distance Estimation

**Yingwei Li**[*] **Tiffany Chen**[†] **Maya Kabkab**[†] **Ruichi Yu**
**Longlong Jing** **Yurong You**[*] **Hang Zhao**[*]

Waymo LLC
{ywli, yuhanc, kabkabm}@waymo.com

## Abstract

Estimating the distance of objects is a safety-critical task for autonomous driving. Focusing on short-range objects, existing methods and datasets neglect the equally important long-range objects. In this paper, we introduce a challenging and under-explored task, which we refer to as Long-Range Distance Estimation, as well as two datasets to validate new methods developed for this task. We then propose R4D, the first framework to accurately estimate the distance of long-range objects by using references with known distances in the scene. Drawing inspiration from human perception, R4D builds a graph by connecting a target object to all references. An edge in the graph encodes the relative distance information between a pair of target and reference objects. An attention module is then used to weigh the importance of reference objects and combine them into one target object distance prediction. Experiments on the two proposed datasets demonstrate the effectiveness and robustness of R4D by showing significant improvements compared to existing baselines. We're looking to make the proposed dataset, Waymo Open Dataset - Long-Range Labels, available publicly, at waymo.com/open/download.

## 1 Introduction

Estimating the distances of objects from the vehicle is crucial for several autonomous driving tasks, including lane changing, route planning, speed adjustment, collision avoidance, to name a few. Although existing methods and datasets focus on short-range objects, knowing the distance of long-range objects — objects beyond a typical LiDAR range of ~80 meters (as shown in Figure 1) — is necessary for freeway driving, heavy-duty truck driving, and wet road driving. Based on the US Department of Transportation (Blanco & Hankey, 2005), on rural highways with a standard speed limit of 65 miles/h, it takes ~145 meters for a passenger vehicle to come to a complete stop in an emergency, greatly exceeding the typical LiDAR sensing range. The required stopping distance grows significantly with a heavy load truck or in bad road conditions such as snow, ice, or rain. For example, the stopping distance increases from 145 meters to 183 meters and 278 meters for trucking and wet road driving, respec-

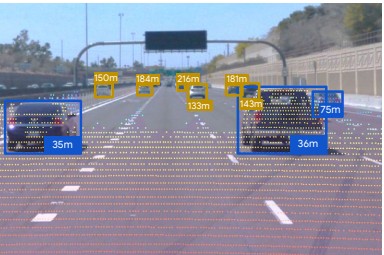

Figure 1: Most long-range **vehicles** (target objects) in this view are beyond **LiDAR** range, but are still important for safety. R4D estimates the distance of **target objects** by using **reference objects** with known distances (*e.g.*, LiDAR detections).

tively (Administration, 2016; Blanco & Hankey, 2005). In addition, given that harsh and sudden breaking on freeways is unsafe, it remains critical to estimate the distance of objects beyond the minimum required stopping distance in order to provide enough time for a gradual slow-down or lane change. Therefore, to allow sufficient time for an appropriate reaction and to ensure safety, autonomous driving systems are required to estimate the distance to long-range objects.

We refer to this critical but underexplored task as *Long-Range Distance Estimation*. Concretely, given the short-range LiDAR signal and the camera image, the output of this task is the distances of

---

[*]Work done while at Waymo LLC. [†]Equal contribution.

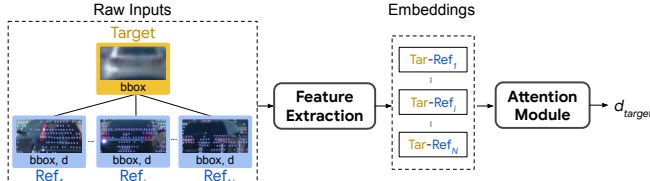

Figure 2: R4D overall architecture for training and prediction. Target and references are represented as a graph and the embeddings are fed to an attention module to weigh different references and aggregate results into a final distance estimate.

long-range objects (beyond LiDAR range). Following existing conventions, the *distance* is measured between the camera and the object center along the camera's optical axis (Gökçe et al., 2015; Haseeb et al., 2018; Zhu & Fang, 2019). For this new task, we introduce two datasets, the *Pseudo Long-Range KITTI Dataset* and the *Waymo Open Dataset - Long-Range Labels*. Since the KITTI (Geiger et al., 2013) dataset does not provide ground-truth distance for long-range objects, the Pseudo Long-Range KITTI dataset is a derived dataset which removes LiDAR points beyond 40 meters and considers all objects beyond 40 meters as long-range. More importantly, we build a new large scale dataset on top of the Waymo Open Dataset which contains annotated *real* long-range objects (with distances ranging from 80 to 300 meters). *We're looking to make this dataset available publicly at* waymo.com/open/download. In summary, while the two datasets defer in their definition of "long-range" (either beyond 40 or 80 meters), both include LiDAR points, camera images, and distance labels for long-range objects.

Neither LiDAR nor camera alone solves this long-range distance estimation problem to the desired accuracy. Most existing *LiDAR* technologies do not meet long-range sensing requirements. According to the Waymo and KITTI self-driving car datasets, the maximum LiDAR range is only around 80 meters, which falls short of the range required for the aforementioned scenarios. Even though some advanced LiDAR systems claim to achieve a longer sensing range, *e.g.*, Waymo's 5th generation LiDAR system (Jeyachandran, 2020) and Velodyne Alpha Prime[TM] (VelodyneLidar, 2021) which reach up to 300 meters, LiDAR points are sparse at long-range and, thus, more likely to be occluded. Therefore, LiDAR alone is not enough to cover all safety-critical self-driving car applications.

*Cameras*, on the other hand, sense objects at a longer range and capture rich semantic information such as object appearance, geometry, and contextual hints. However, they do not, in and of themselves, provide depth information. Classical geometry-based algorithms could estimate the distance of canonical objects (*e.g.*, sedans, trucks) based on their pixel sizes in the camera image (Criminisi et al., 2000; Tuohy et al., 2010; Gökçe et al., 2015; Haseeb et al., 2018; Qi et al., 2019). However, these approaches yield inaccurate results on long-range objects due to errors in size estimation. Appearance-based methods (Song et al., 2020; Zhu & Fang, 2019) give unsatisfactory results on long-range objects. Such methods rely on a single appearance cue to estimate the distance, overlooking context or other relevant signals in the scene. At long-range, objects are visually small resulting in less informative appearance features. Although neither LiDAR nor camera alone can solve long-range distance estimation, these two signals provide complementary cues for this task.

In this paper, we propose R4D, a method to utilize reference objects with known and accurate distances for long-range distance estimation (R4D). The main motivation behind our approach lies in theories of human perception from cognitive science: humans often estimate the distance of an object relative to other *reference* points or objects (Granrud et al., 1984). Specifically, we train a model to localize a long-range object (*target*) given *references* with known distances. We represent the target object and references as a *graph*. As shown in Figure 2, we define objects as nodes, with edges connecting the target to the reference objects. The reference information is propagated to the long-range target object by extracting target-reference (Tar-Ref) embeddings. R4D then feeds all target-reference embeddings to an attention module which fuses information from different references, by weighing their relative importance and combining them into one distance prediction.

Experiments on two proposed datasets show that R4D significantly improves the distance estimation and achieves state-of-the-art long-range localization results. For example, on the *Waymo Open Dataset - Long-Range Labels*, with R4D, the distances of 8.9% more vehicles (from 53.4% to 62.3%) are predicted with a relative distance error below 10%. By conducting experiments on images captured at different times of the day, *e.g.*, train on daytime and test on nighttime images, R4D also shows stronger robustness against domain changes.

To summarize, our contributions are three-fold. (1) We propose a critical but underexplored task Long-Range Distance Estimation. (2) We present two datasets, the Pseudo Long-Range

KITTI Dataset and Waymo Open Dataset - Long-Range Labels. To facilitate future research, we are looking to make Waymo Open Dataset - Long-Range Labels available publicly at waymo.com/open/download. (3) We develop R4D, the first framework to accurately estimate the distance of long-range objects by using references with known distances.

## 2 Related Work

In this section, we discuss the related tasks and the detailed design choices of our proposed approach.

**Monocular distance estimation.** Estimating the distance of objects from an RGB image (*i.e.*, monocular distance estimation) is a popular computer vision topic. More than a decade ago, researchers designed geometry-based algorithms to estimate object distances. For example, Tuohy et al. (2010) use inverted perspective mapping (IPM) to convert an image to the bird's eye view for estimating distances. Learning-based methods were later proposed (Qi et al., 2019; Song et al., 2020). For example, SVR (Gökçe et al., 2015) and DisNet (Haseeb et al., 2018) take the pixel height and width of an object as input and estimate object distance by using support vector regression (Drucker et al., 1997) and multi-layer perceptron (MLP). However, such methods are subject to errors in object size estimation (Zhu & Fang, 2019), which can be even more pronounced for long-range objects. Zhu & Fang (2019) developed an end-to-end distance estimator with a Convolutional Neural Network (CNN) feature extractor. Specifically, given an RGB image and object bounding boxes, the model extracts per-object embeddings with a CNN and ROI pooling operator (Girshick, 2015), and then predicts a distance for each object. As previously mentioned, this method relies heavily on object appearance, which is a less informative cue for long-range objects.

Another related research topic is monocular per-pixel depth estimation. Eigen et al. (2014); Garg et al. (2016); Godard et al. (2019); Lee & Kim (2019); Liu et al. (2015); Shu et al. (2020) propose to predict dense depth maps given RGB images. For example, Lee & Kim (2019) generate multi-resolution relative depth maps to construct the final depth map. However, these methods can be resource-intensive and, therefore, difficult to incorporate within a latency-sensitive system such as autonomous driving (Zhu & Fang, 2019). In addition, it is non-trivial to translate a depth map into per-object distance estimates due to occlusions, looseness of bounding boxes, *etc*.

**Long-range depth sensing.** Other approaches such as stereo cameras and Radar have been used to sense objects at long-range. These approaches are discussed in Section K due to space limitations.

**Monocular 3D object detection.** Monocular 3D object detection aims to detect all objects in an image by predicting a 3D bounding box for each of them. In Section K, we discuss why *our task focuses on estimating the distance of long-range objects*.

## 3 Methodology

Motivated by the observation that humans estimate the distance of an object relative to other references (Granrud et al., 1984), we propose R4D, a method utilizing Reference objects For long-range Distance estimation. References can be any combination of objects or points with known and ac-

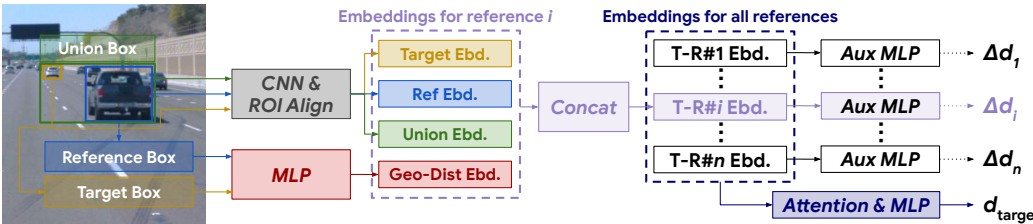

Figure 3: The detailed architecture for training and prediction. We use a CNN to extract a feature map from the input image. Given target, reference, and union bounding boxes, an ROIAlign is used to crop their corresponding embeddings from the feature map. Given the center and size of the target and reference boxes, we extract geo-distance embeddings using a MLP. We refer to "Attention & MLP" and its output as the *absolute distance head*, and "Aux MLP" and its output as the *relative distance head*. *Relative heads* are only used during training. $\Delta d_i$ is the relative distance between the target and the $i^{\text{th}}$ reference object.

curate distances to the autonomous vehicle, such as LiDAR detections, objects detected by other sensors, map features, *etc*. We represent the target object and references as a *graph* shown in the "Raw Inputs" dashed box in Figure 2. The target object and its references are nodes in this graph. Edges connect a target to its reference objects to encode their pairwise relationships.

The detailed architecture is illustrated in Figure 3. To model the target-reference pairwise relationship, we propose to extract *union embeddings* and *geo-distance embeddings* which encode visual and geometry relationships, respectively (Section 3.1). Then, inspired by the intuition that reference objects are not equally important, we introduce an attention module in Section 3.2 to selectively aggregate pairwise relationships. Finally, R4D is trained with an auxiliary supervision: the *relative distances* between the target and its references, as explained in Section 3.3.

In our setup, and without loss of generality, we use a monocular camera as the main sensor for detecting long-range target objects, and adopt LiDAR detected short-range objects as references. It is worth noting that R4D is not specifically designed for LiDAR or monocular images and is readily extended to other sensors and references.

## 3.1 MODELING PAIRWISE RELATIONSHIPS

As previously mentioned, to better estimate the distance of a target object, we propose to model the pairwise relationships between it and other reference objects. For every pair of reference-target objects, we extract per-object feature embeddings and pairwise embeddings, as shown in Figure 3. In particular, pairwise embeddings include *union* embeddings and *geo-distance* embeddings which provide visual and geometry cues, respectively.

**Union embeddings.** Visual cues for modeling pairwise relationships should be based on the target object, the reference object, as well as the scene between the two objects. We therefore propose a simple approach to extract feature embeddings from the union bounding box which covers both objects. Figure 3 shows an example of a union bounding box highlighted in green. This is similar to Yao et al. (2018), where union bounding boxes were used to extract features for image captioning.

**Geo-distance embeddings.** To provide geometry cues, we form an input using the following:

- 2D bounding box *center coordinates* of target and reference objects, and the relative position shift between them.
- 2D bounding box *size* of target and reference objects, and the relative scale between them.
- The *distance of the reference object* as provided by a LiDAR 3D object detector.

We, then, feed this input into a multi-layer perceptron to generate the geo-distance embedding.

Given the target, reference, union, and geo-distance embeddings, we concatenate them to form a final target-reference embedding which models the relationship between a pair of target and reference objects. We analyze and discuss the importance of these embeddings in Section 5.3.

## 3.2 ATTENTION-BASED INFORMATION AGGREGATION

Pairs of target-reference embeddings should be combined to estimate the distance of a given target object. A simple approach is to average them over all references of the same target object. However, intuitively, and as supported by our experimental results, reference objects are not equally important. For example, a car in the bottom left of the image is potentially less helpful when localizing a faraway car lying in the top right corner of the image.

In order to guide our model to focus on the most important reference objects, we introduce an attention-based module. As shown in Figure 4, we follow VectorNet (Gao et al., 2020) to construct embeddings with local and global information. Specifically, we use MLP and average pooling to extract a global embedding (shown in yellow in Figure 4). The global embedding is, then, concatenated with the original

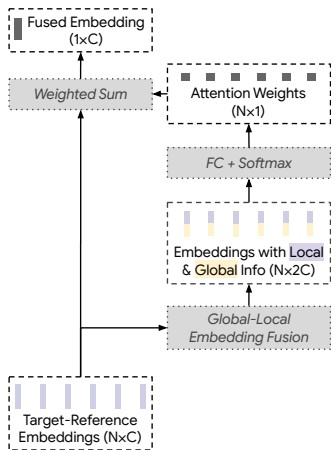

Figure 4: Attention module to fuse target-reference embeddings.

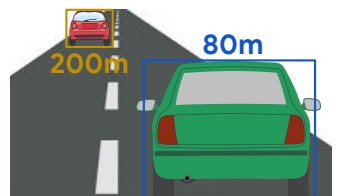
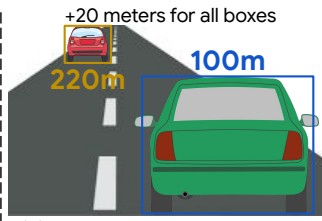

Figure 5: Data distribution of the anonymous long-range dataset.

Figure 6: An example of our distance augmentation for encouraging learning pairwise relationships.

target-reference embeddings (blue in Figure 4). Given these global-local embeddings, the attention module predicts and normalizes importance weights for each reference using a fully connected layer followed by a softmax layer. It, finally, uses these weights to fuse the original target-reference embeddings, in a weighted average manner, resulting in one final embedding.

### 3.3 RELATIVE DISTANCE SUPERVISION

Predicting the target distance from the target embedding is more straightforward than from other indirect cues such as reference embeddings. To encourage the model to learn the pairwise relationships between target and reference objects, instead of shortcut cues (Geirhos et al., 2020; Li et al., 2021; Wang et al., 2020), we provide additional supervision. The design of this additional supervision is akin to the residual representation, which is widely used for computer vision to help the optimization procedure (Briggs et al., 2000; He et al., 2016; Szeliski, 1990; 2006). Specifically, during the training stage, we add a relative (or residual) distance head for each target-reference embedding. The *relative* distance $\Delta d$ between the target object and the reference object is given by $\Delta d = d_t - d_r$, where $d_t$ (resp., $d_r$) is the distance of the target (resp., reference) object.

## 4 DATASETS AND EVALUATION METRICS

To evaluate methods on the new task of long-range distance estimation, we generate the *Pseudo Long-Range KITTI Dataset* and introduce the *Waymo Open Dataset - Long-Range Labels*.

### 4.1 PSEUDO LONG-RANGE KITTI DATASET

The original KITTI Dataset (Geiger et al., 2013) contains RGB images, LiDAR point clouds, as well as 2D and 3D bounding boxes and other per-pixel annotations. With these rich annotations, benchmarks for self-driving tasks were developed on the KITTI Dataset, including scene flow estimation, depth estimation, detection, and segmentation.

Even though the KITTI Dataset provides LiDAR point clouds, it does not provide distance annotations beyond LiDAR range. We hereby present the *Pseudo Long-Range KITTI Dataset*, which is derived from the original KITTI Dataset. By assuming that LiDAR has an effective sensing range of 40 meters only, we remove LiDAR points which are further away. The objects lying beyond this effective sensing range are considered as long-range target objects, while the distance of other objects are known and provided as inputs. It is worth noting that there is no overlap between the aforementioned target objects and the objects with known distance. Following the convention (Shi et al., 2019), the original training data is split into two subsets for training and validation, respectively. Images which do not contain any long-range objects are removed from the dataset. As such, the Pseudo Long-Range KITTI Dataset contains 2,181 images with 4,233 vehicles, and 2,340 images with 4,033 vehicles in the training and validation sets, respectively. This derived dataset is relatively small-scale and more importantly does not include real long-range objects.

### 4.2 WAYMO OPEN DATASET - LONG-RANGE LABELS

Given the aforementioned limitations of the Pseudo Long-range KITTI Dataset, we build a new long-range dataset which is built on top of the Waymo Open Dataset and includes vehicle distances up to 300 meters from the autonomous vehicle. The labels are obtained as follows. First, we create 3D boxes for objects in the LiDAR range. Then, we use Radar derived signals for rough positioning

| Method | LiDAR | < 5%↑ | < 10%↑ | < 15%↑ | Abs Rel↓ | Sq Rel↓ | RMSE↓ | RMSE$_{log}$↓ |
|---|---|---|---|---|---|---|---|---|
| SVR | ✗ | 29.9% | 51.6% | 65.4% | 12.8% | 5.43 | 31.4 | 0.210 |
| DisNet | ✗ | 31.3% | 52.4% | 66.5% | 12.5% | 5.28 | 31.3 | 0.207 |
| Zhu & Fang (2019) | ✗ | 31.2% | 53.4% | 68.7% | 11.5% | 4.43 | 29.4 | 0.187 |
| Zhu & Fang (2019) | ✓ | 34.1% | 56.4% | 70.5% | 11.3% | 4.78 | 31.3 | 0.194 |
| R4D-NA (**ours**) | ✓ | 35.9% | 61.7% | 76.4% | 10.1% | 3.68 | 27.0 | 0.168 |
| R4D (**ours**) | ✓ | **38.5%** | **62.3%** | **77.1%** | **9.8%** | **3.60** | **26.7** | **0.167** |

Table 1: Results on the Waymo Open Dataset - Long-Range Labels. Compared to Zhu & Fang (2019), R4D achieves performance gains on all evaluation metrics. R4D also outperforms SVR (Gökçe et al., 2015), and DisNet (Haseeb et al., 2018). R4D-NA refers to No-Attention R4D.

and extend LiDAR boxes to long-range. Due to the low resolution of Radar, labelers use smooth object trajectory constraints to make sure the boxes are consistent over time and space. Finally, given the 3D boxes for long-range objects, we compute their distances to the autonomous vehicle. In total, as shown in Figure 5, we obtain 49,056 training images with 187,938 long-range vehicles, and 3,578 validation images containing 10,483 long-range vehicles. These images span different times of day, including dawn, day, dusk, and night time. Given its size and the real long-range annotations, we use this dataset by default.

### 4.3 EVALUATION METRICS

We follow the evaluation metrics used in dense depth and per-object distance estimation (Eigen et al., 2014; Liu et al., 2015; Garg et al., 2016; Zhu & Fang, 2019; Shu et al., 2020; Zhang et al., 2020), namely: 1) the percentage of objects with relative distance error below a certain threshold ($< 5\%$↑, $< 10\%$↑, $< 15\%$↑), 2) the absolute relative difference (Abs Rel↓), 3) the squared relative difference (Sq Rel↓), 4) the root of the mean squared error (RMSE↓), and 5) the root of the mean squared error computed from the logs of the predicted and ground-truth distances (RMSE$_{log}$↓). Specifically, given ground-truth distances $d^*$ and predicted distances $d$, we compute the metrics as follows:

$$
\begin{aligned}
\text{Abs Rel↓} &= \text{Average}\left(|d - d^*|/d^*\right), & \text{Sq Rel↓} &= \text{Average}\left((d - d^*)^2/d^*\right), \\
\text{RMSE↓} &= \sqrt{\text{Average}\left((d - d^*)^2\right)}, & \text{RMSE}_{log}\text{↓} &= \sqrt{\text{Average}\left((\log d - \log d^*)^2\right)}.
\end{aligned}
\tag{1}
$$

Note that ↑ and ↓ indicate whether a larger or smaller value of the corresponding metric indicates a better performance. We only evaluate performance on long-range objects.

## 5 EXPERIMENTS

### 5.1 IMPLEMENTATION DETAILS

Detailed design choices are outlined in Section A, while this section focuses on introducing a critical component used during training: distance augmentation.

**Distance augmentation.** Distance augmentation is designed to encourage R4D to learn from pairwise relationships and prevent it from exclusively using a single shortcut cue (namely, the target embeddings). We guide the model to focus on the relative distance between the reference and target objects by emphasizing the correlation between the pairwise embeddings and the distance prediction. Specifically, we keep the relative distance fixed, perturb the reference distance, and expect the model to predict the target distance with the same perturbation. For example, as shown in Figure 6 (a), the relative distance between the target and reference object is 120 meters and given a reference distance of 80 meters, the model should predict a target distance of 200 meters (= 80 meters + 120 meters). As shown in Figure 6 (b), when the provided reference distance is perturbed

| Method | LiDAR | < 5%↑ | < 10%↑ | < 15%↑ | Abs Rel↓ | Sq Rel↓ | RMSE↓ | RMSE$_{log}$↓ |
|---|---|---|---|---|---|---|---|---|
| SVR | ✗ | 39.1% | 65.9% | 79.0% | 10.2% | 1.38 | 9.5 | 0.166 |
| DisNet | ✗ | 37.1% | 65.0% | 77.7% | 10.6% | 1.55 | 10.4 | 0.181 |
| Zhu & Fang (2019) | ✗ | 39.4% | 65.8% | 80.2% | 8.7% | 0.88 | 7.7 | 0.131 |
| Zhu & Fang (2019) | ✓ | 41.1% | 66.5% | 78.0% | 8.9% | 0.97 | 8.1 | 0.136 |
| R4D (**ours**) | ✓ | **46.3%** | **72.5%** | **83.9%** | **7.5%** | **0.68** | **6.8** | **0.112** |

Table 2: Results on the pseudo long-range KITTI dataset. R4D outperforms SVR (Gökçe et al., 2015), DisNet (Haseeb et al., 2018), and Zhu & Fang (2019).

| Tgt | Ref | Uni | G-D | $< 5\%\uparrow$ | $< 10\%\uparrow$ | $< 15\%\uparrow$ | Abs Rel$\downarrow$ | Sq Rel$\downarrow$ | RMSE$\downarrow$ | RMSE$_{\log}\downarrow$ |
|-----|-----|-----|-----|------|------|------|------|------|------|------|
| ✓ | ✗ | ✗ | ✗ | 31.2% | 53.4% | 68.7% | 11.5% | 4.43 | 29.4 | 0.187 |
| ✓ | ✓ | ✗ | ✗ | 31.5% | 60.8% | 75.3% | 11.2% | 4.30 | 29.2 | 0.183 |
| ✓ | ✓ | ✓ | ✗ | 32.5% | 55.2% | 69.7% | 11.2% | 4.42 | 29.7 | 0.188 |
| ✓ | ✓ | ✗ | ✓ | 36.9% | 60.8% | 75.3% | 10.2% | 3.88 | 27.8 | 0.175 |
| ✓ | ✓ | ✓ | ✓ | **38.5%** | **62.3%** | **77.1%** | **9.8%** | **3.60** | **26.7** | **0.167** |

Table 3: The target embeddings (Tgt) and the reference embeddings (Ref) alone are not enough for modeling pairwise relationships. The union embeddings (Uni) and geo-distance embeddings (G-D) are also necessary components of R4D.

(by, say, 20 meters), the model should predict 220 meters (= 100 meters + 120 meters) as the target distance, in order to maintain the correct relative distance of 120 meters. This helps the model not overfit to appearance cues provided by the target object and be more robust to slight changes in camera parameters or field-of-view. Similar augmentation techniques have been successfully used to discourage shortcut learning in other deep learning tasks (Li et al., 2021). During training, the distance label perturbation is sampled from a Gaussian distribution $X \sim \mathcal{N}(\mu, \sigma^2)$ with $\mu = 0$ and $\sigma = 200$ (See Section B for details). For the Pseudo Long-Range KITTI Dataset, we use $\sigma = 50$.

## 5.2 LONG-RANGE DISTANCE ESTIMATION

Tables 1 and 2 show the distance estimation results on the Waymo Open Dataset - Long-Range Labels and the Pseudo long-range KITTI Dataset. On both datasets, R4D significantly outperforms prior monocular distance estimation solutions. For example, compared to Zhu & Fang (2019), R4D significantly improves the performance on the Waymo Open Dataset - Long-Range Labels by 7.3%, 8.9%, and 8.4% on the $< 5\%\uparrow$, $< 10\%\uparrow$, and $< 15\%\uparrow$ metrics, respectively. We observe similar trends on the Pseudo Long-Range KITTI Dataset: these improve by 6.9%, 6.7%, and 3.7%, respectively. These results show the effectiveness of R4D.

To further illustrate that R4D effectively uses references in LiDAR range for long-range distance estimation, we design another simple baseline to use LiDAR as a reference. We augment the input camera image (RGB) with a channel containing per-pixel projected LiDAR distance values (D) (shown in Figure 1). Then, given an RGB+D image, and detected bounding boxes, the model predicts a distance value for each target object. We modify the input of our baseline, Zhu & Fang (2019), to include the distance channel (D), in addition to the RGB image. However, as the results show in Tables 1 and 2 (row 4), this approach achieves similar performance as when the camera image (RGB) is the sole input. In addition, in Figure 7, we show the distance estimation results of R4D as well as Zhu & Fang (2019) with RGB and RGB+D inputs, broken-down over different ground-truth distance ranges. We note that the addition of the LiDAR distance channel only improves the performance of objects slightly beyond LiDAR range, but fails at extrapolating distance values for localizing extremely long-range objects. In contrast, R4D offers consistent performance gains compared to Zhu & Fang (2019) across all distances. This results demonstrates the importance of explicitly modeling object relationships (the strategy of R4D), compared to the simpler baseline which has access to the same LiDAR sensor data.

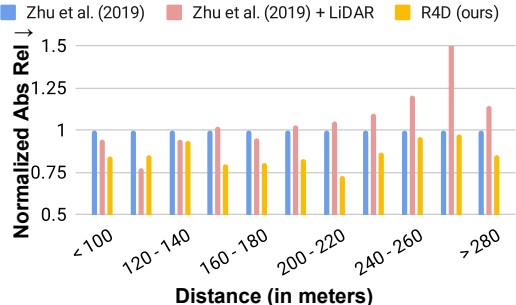

Figure 7: Comparison between Zhu & Fang (2019) with and without LiDAR information. We show the Abs Rel$\downarrow$ metric (normalized by the blue bars) for different ground-truth distance ranges. Results show that the simpler way of fusing short-range LiDAR signals marginally improves the performance on objects slightly beyond LiDAR range but hurts very long-range objects.

## 5.3 ABLATION STUDIES

**Attention-based information aggregation.** As discussed in Section 3.2, given a fixed target, R4D aggregates all target-reference embeddings using an attention-based module and predicts a weight for each reference. As shown in Table 1, compared to treating all references equally and taking the average of all target-reference embeddings (R4D-NA), the attention module improves the performance across all metrics, *e.g.*, <5%↑ is improved by 2.6%.

| Method | Abs. | Rel. | #Ref | DA | < 5%↑ | < 10%↑ | < 15%↑ | Abs Rel↓ | Sq Rel↓ | RMSE↓ | RMSE$_{log}$↓ |
|---|---|---|---|---|---|---|---|---|---|---|---|
| Zhu & Fang (2019) | ✓ | ✗ | Zero | ✗ | 32.9% | 55.6% | 69.9% | 11.2% | 4.36 | 29.3 | 0.187 |
| Zhu & Fang (2019) | ✓ | ✗ | Zero | ✓ | 32.3% | 54.3% | 69.4% | 11.4% | 4.37 | 29.2 | 0.186 |
| R4D-Relative | ✗ | ✓ | One | ✗ | 34.4% | 58.8% | 73.8% | 10.6% | 4.00 | 28.1 | 0.178 |
| R4D-Relative | ✗ | ✓ | One | ✓ | 34.9% | 57.4% | 73.3% | 10.6% | 3.93 | 27.8 | 0.174 |
| R4D-One-Ref | ✓ | ✓ | One | ✗ | 37.5% | 59.9% | 74.3% | 10.4% | 3.92 | 27.7 | 0.176 |
| R4D-One-Ref | ✓ | ✓ | One | ✓ | 37.5% | 59.5% | 73.8% | 10.3% | 3.87 | 27.6 | 0.175 |
| R4D-NoAuxLoss | ✓ | ✗ | All | ✗ | 29.9% | 52.1% | 69.7% | 11.6% | 4.67 | 30.4 | 0.195 |
| R4D-NoAuxLoss | ✓ | ✗ | All | ✓ | **37.9%** | 60.9% | 74.7% | 10.2% | 3.88 | 27.9 | 0.174 |
| R4D | ✓ | ✓ | All | ✗ | 32.0% | 54.1% | 70.9% | 11.1% | 4.30 | 29.3 | 0.187 |
| R4D | ✓ | ✓ | All | ✓ | 37.5% | **63.3%** | **78.6%** | **9.8%** | **3.64** | **26.9** | **0.169** |

Table 4: Ablation studies on the proposed pairwise relationship modeling and distance augmentation (DA). Results in this table are slightly different from those in Tables 1 and 3 because, here, we remove images which do not contain any reference objects. This is in order to report results on R4D-Relative, which requires at least one reference object (see Section 5.3 for details).

In order to inspect the attention module output, we visualize the learnt reference weights. We highlight the reference object with the largest weight in Figure 8. We find that results match our intuition. For example, the most important references usually share the same lane and pose as the target. In addition, the best reference is usually the one closest to the target object.

**Efficacy of feature embeddings.** In this section, we perform ablation experiments on the proposed feature embeddings (Section 3.1) to show their importance. The first two rows in Table 3 show that, when only using the target and reference embeddings, the performance is sub-optimal. This is likely due to the fact that these embeddings fail to capture pairwise relationships and only encode individual appearance, pose, and geometry information. Incorporating the geo-distance embeddings (row 4) allows the model to learn from the relative position and geometry between target and reference objects, resulting in the largest performance improvement. Finally, adding the union embeddings (row 5), which encode relative appearance cues, further improves the performance (*e.g.*, < 5%↑ improves by 1.6%). Note that the union embeddings are only helpful when used together with the geo-distance embeddings. This is because the union bounding box may include multiple potential objects, and a precise position is critical to pinpoint the reference object location.

**Benefits of modeling pairwise relationships.** R4D's main performance improvements stem from modeling pairwise relationships between target and reference objects. In this section, we show how incorporating references helps. We study the effect of removing the relative distance supervision (*i.e.*, the "Aux MLP" head in Figure 3) or the absolute distance supervision. When only the relative distance head is enabled, we use its output during inference and the target distance is computed by adding the relative distance to the known reference distance (we exclude images without references).

As shown in Table 4, randomly choosing one reference (row 3) and predicting the distance using relative distance achieves a < 10%↑ of 58.8%, outperforming the baseline (Zhu & Fang, 2019) which directly predicts the absolute distance (row 1 with a < 10%↑ of 55.6%). Combining the absolute and relative heads during training (row 5) further improves the < 10%↑ metric by 4.3%

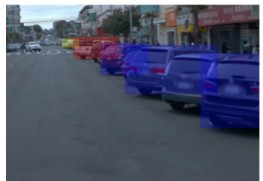 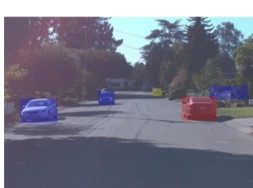 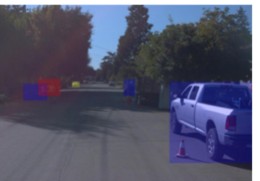 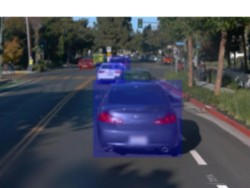

Figure 8: Visualizing the learnt reference weights. Targets are highlighted in yellow. References with the largest attention weights are highlighted in red. Other references are colored in blue. We observe that the most important references are often close to or in the same lane as the target, and share a similar pose as the target. Best viewed zoomed in. More images are shown in Section F.

| Valid. Set | Method | < 15%↑ | Abs Rel↓ | Sq Rel↓ | RMSE↓ | RMSE_log↓ |
|---|---|---|---|---|---|---|
| Day | Zhu & Fang (2019) | 72.2% | 11.2% | 4.52 | 30.3 | 0.190 |
| | R4D (**ours**) | 77.0% (+4.8%) | 9.6% (-1.6%) | 3.71 (-0.81) | 27.8 (-2.5) | 0.171 (-0.019) |
| Dawn & Dusk | Zhu & Fang (2019) | 63.8% | 13.6% | 6.48 | 36.4 | 0.233 |
| | R4D (**ours**) | 70.0% (+6.2%) | 11.3% (-2.3%) | 4.19 (-2.29) | 28.5 (-7.9) | 0.177 (-0.056) |
| Night | Zhu & Fang (2019) | 36.8% | 23.3% | 12.5 | 46.8 | 0.344 |
| | R4D (**ours**) | 49.1% (**+12.3%**) | 17.7% (**-5.6%**) | 8.2 (**-4.38**) | 38.4 (**-8.4**) | 0.267 (**-0.077**) |

Table 5: Robustness on out-of-distribution data. Both R4D and the baseline (Zhu & Fang, 2019) are trained on daytime images only, and evaluated on three test sets collected during different times of day. The performance improvement of R4D increases when the domain gap is larger.

(achieving a $< 10\%\uparrow$ metric of 59.9%). Finally, this improvement is the largest when multiple references are aggregated using the attention module (row 8), resulting in a $< 10\%\uparrow$ of 63.3%.

**Distance augmentation.** As described in Section 5.1, we propose the distance augmentation strategy to encourage R4D to learn better pairwise relationships during training. Table 4 shows the results from ablation experiments related to distance augmentation. The distance augmentation is only critical for our final model R4D, which (1) includes both absolute and relative distance prediction heads, and (2) considers *multiple* reference objects. Without distance augmentation (row 7), the model does not learn pairwise relationships, and the performance of R4D's absolute distance head drops to the baseline's performance (row 1). In fact, when at most one reference is used during training (Zhu & Fang (2019), R4D-Relative, and R4D-One-Ref), the proposed distance augmentation marginally affects the performance. Our intuition is that the one- or zero-reference cases require less supervision and data diversity due to the simpler input and model structures.

## 5.4 Robustness on Out-Of-Distribution Data

For safety-critical applications, such as autonomous driving, the ability to handle out-of-distribution (OOD) data is crucial (Vyas et al., 2018). We expect R4D to be more robust on OOD data because it captures contextual information, in addition to appearance and geometry cues. In contrast, baseline methods (Zhu & Fang, 2019) use a single prediction cue, thus rendering them more sensitive to appearance changes. To validate the robustness on OOD data, we train our model solely on images captured during daytime, but evaluate on three different times of day, namely: daytime, dawn/dusk, and nighttime. Specifically, we split the Waymo Open Dataset - Long-Range Labels into four disjoint sets: *train-day* (used for training), and *val-day*, *val-dawn&dusk*, *val-night* (used for validation). The results of this experiment are shown in Table 5. Compared to the baseline method (Zhu & Fang, 2019), R4D achieves a larger improvement when the domain gap between the training and validation sets is larger. For example, when evaluating on daytime (small domain gap) and nighttime (large domain gap), R4D achieves an Abs Rel↓ metric improvement of 1.6% and 5.6%, respectively.

## 6 Conclusion

In this paper, we presented an underexplored task, long-range distance estimation, as well as two datasets for it. We then introduced R4D, a novel approach to address this problem. Our method can accurately localize long-range objects using references with known distances in the scene. Experiments on two proposed datasets demonstrate that it significantly outperforms prior solutions and achieves state of the art long-range distance estimation performance. It should also be noted that R4D is not specifically designed for LiDAR or monocular images and is readily extended to other sensors and references, such as features from high-definition maps. It can also be used to incorporate temporal information by fusing past object detections as references, which is a direction we hope to explore to further improve the long-range distance estimation performance.

**Acknowledgement.** We want to thank Tom Ouyang, Alper Ayvaci, Ching-Hui Chen, Henrik Kretzschmar, Zhinan Xu, Charles R. Qi, Jiayuan Gu, Fan Yang, and Dragomir Anguelov for their help on this project.

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

## A    IMPLEMENTATION DETAILS

We use ResNet-50 (He et al., 2016) followed by an ROIAlign layer (He et al., 2017) to extract target, reference, and union embeddings. We then apply a fully connected layer to each embedding to reduce the feature dimension to 1,024. We use a three-layer perceptron to extract geo-distance embeddings. Each layer contains a single fully connected layer followed by layer normalization (Ba et al., 2016) and ReLU (Nair & Hinton, 2010). For the attention module, we use the two-layer subgraph propagation operation mentioned in VectorNet (Gao et al., 2020) as the Global-Local Embedding Fusion depicted in Figure 4 in the main submission, and replace the max pooling layer with average pooling. The reference objects are detected using a multi-view fusion 3D object detector (Zhou et al., 2020) with 72.1 3D AP (L1). When there is no reference object in an image (0.6% of the images), we disable the relative distance supervision. We use smooth L1 (Huber, 1992) as the loss function (Zhu & Fang, 2019). We train R4D on an 8-core TPU with a total batch size of 32 images for 24 epochs. The learning rate is initially set to 0.0005 and then decays by a factor of 10 at the 16th and 22nd epochs. We warm up (Goyal et al., 2017) the learning rate for 1,800 iterations. We initialize the weights from a ResNet-50 model pretrained on ImageNet (He et al., 2016). For the Pseudo Long-Range KITTI Dataset, we train over 150 epochs with an initial learning rate of 0.005 and cosine learning rate decay (Loshchilov & Hutter, 2017).

| DistAug | $\sigma$ | $< 5\%$↑ | $< 10\%$↑ | $< 15\%$↑ | Abs Rel↓ | Sq Rel↓ | RMSE↓ | RMSE$_{log}$↓ |
|---------|----------|----------|-----------|-----------|----------|---------|-------|---------------|
| ✗ | N/A | 32.0% | 54.1% | 70.9% | 11.1% | 4.30 | 29.3 | 0.187 |
| ✓ | 1 | 32.8% | 56.5% | 71.9% | 11.0% | 4.32 | 29.2 | 0.186 |
| ✓ | 2 | 32.6% | 54.9% | 71.6% | 10.9% | 4.13 | 28.7 | 0.182 |
| ✓ | 5 | 32.4% | 55.0% | 71.0% | 11.2% | 4.44 | 29.8 | 0.189 |
| ✓ | 10 | 33.9% | 56.8% | 73.3% | 10.7% | 4.18 | 29.0 | 0.183 |
| ✓ | 20 | 35.7% | 58.1% | 73.4% | 10.3% | 3.89 | 28.0 | 0.175 |
| ✓ | 50 | 36.6% | 59.5% | 75.1% | 10.2% | 3.91 | 28.1 | 0.175 |
| ✓ | 100 | 34.6% | 58.3% | 75.0% | 10.4% | 3.91 | 28.0 | 0.176 |
| ✓ | 150 | 37.5% | 62.1% | 76.6% | 9.9% | **3.61** | **26.8** | **0.167** |
| ✓ | 200 | 37.5% | **63.3%** | **78.6%** | **9.8%** | 3.64 | 26.9 | 0.169 |
| ✓ | 250 | **38.3%** | 61.0% | 75.2% | 10.1% | 3.79 | 27.5 | 0.173 |
| ✓ | 300 | 37.2% | 60.7% | 75.6% | 10.1% | 3.77 | 27.4 | 0.171 |

Table 6: Hyper-parameter sensitivity of distance augmentation.

| Method | #Ref | $< 5\%$↑ | $< 10\%$↑ | $< 15\%$↑ | Abs Rel↓ | Sq Rel↓ | RMSE↓ | RMSE$_{log}$↓ | T (ms)↓ |
|--------|------|----------|-----------|-----------|----------|---------|-------|---------------|---------|
| Zhu & Fang (2019) | 0 | 32.0% | 54.1% | 70.9% | 11.1% | 4.30 | 29.3 | 0.187 | **15.0** |
| R4D(**ours**) | 1 | 35.0% | 60.7% | 76.0% | 10.4% | 3.93 | 27.9 | 0.175 | 15.2 |
| R4D(**ours**) | 2 | 37.0% | 62.4% | 77.5% | 10.0% | 3.83 | 27.7 | 0.174 | 15.5 |
| R4D(**ours**) | 5 | 37.2% | 63.3% | 78.4% | **9.8%** | 3.68 | 27.1 | 0.170 | 15.6 |
| R4D(**ours**) | 10 | 37.3% | **63.7%** | 78.5% | **9.8%** | 3.66 | 27.0 | **0.169** | 16.3 |
| R4D(**ours**) | 20 | 37.4% | 63.4% | 78.5% | **9.8%** | **3.64** | **26.9** | **0.169** | 17.0 |
| R4D(**ours**) | 50 | **37.5%** | 63.3% | **78.6%** | **9.8%** | **3.64** | **26.9** | **0.169** | 20.3 |

Table 7: Effect of the maximum number of references (#Ref) on distance estimation performance and inference latency (T).

## B    HYPER-PARAMETER SENSITIVITY: DISTANCE AUGMENTATION

In this section, we show how the performance of R4D varies with the distance augmentation hyper-parameter $\sigma$. In our final model, we perturbed the ground-truth distances of target and reference objects by a Gaussian random variable with distribution $X \sim \mathcal{N}(\mu, \sigma^2)$ with $\mu = 0$ and $\sigma = 200$. In Table 6, we show the performance of our method for different values of $\sigma$ ranging from 1 to 300. We observe that the proposed distance augmentation consistently improves the performance of R4D for all values of $\sigma$. The model reaches its optimal performance when $\sigma$ is around 200.

## C    INFERENCE LATENCY AND NUMBER OF REFERENCES

For all previous results in this paper, we set the maximum number of references to 50. In this section, we show the performance and inference time of our method when we vary the maximum number of references. We draw two conclusions from the results (in Table 7). First, more references lead to better results. For example, we observe that, with a maximum of 2 reference objects, the $< 5\%$↑ metric improves by 2% (from 35.0% to 37.0%) compared to using a single reference object. In addition, when the number of references is large (say, 5), the benefit of using additional reference objects is marginal. For example, we observe that the $< 5\%$↑ metric increases by only 0.3% (from 37.2% to 37.5%) when the maximum number of references is increased from 5 to 50. Finally, and as expected, the latency increases with the number of reference objects. However, this increase is not significant when using a relatively small number of references. For example, when we increase the maximum number of references from 1 to 5, the latency only increases by 0.6 ms, while the $< 10\%$↑ metric improves by 9.2%.

## D    PSEUDO LONG-RANGE NUSCENES DATASET

We further provide another derived pseudo long-rang dataset, *Pseudo Long-Range nuScenes Dataset*, and the experimental results on it. Although we provide two derived pseudo long-range datasets, we still encourage readers to focus more on our Waymo Open Dataset - Long-Range La-

| Method | LiDAR | $< 5\%$↑ | $< 10\%$↑ | $< 15\%$↑ | Abs Rel↓ | Sq Rel↓ | RMSE↓ | RMSE$_{log}$↓ |
|---|---|---|---|---|---|---|---|---|
| SVR | ✗ | 33.0% | 58.0% | 74.0% | 10.9% | 1.56 | 10.7 | 0.166 |
| DisNet | ✗ | 29.5% | 58.6% | 75.0% | 10.7% | 1.46 | 10.5 | 0.162 |
| Zhu & Fang (2019) | ✗ | 40.3% | 66.7% | 80.3% | 8.4% | 0.91 | 8.6 | 0.121 |
| Zhu & Fang (2019) | ✓ | 37.7% | 63.5% | 77.2% | 9.2% | 1.06 | 9.2 | 0.132 |
| R4D (**ours**) | ✓ | **44.2%** | **71.1%** | **84.6%** | **7.6%** | **0.75** | **7.7** | **0.110** |

Table 8: Results on the pseudo long-range nuScenes Dataset. R4D outperforms SVR (Gökçe et al., 2015), DisNet (Haseeb et al., 2018), and Zhu & Fang (2019).

| Method | LiDAR | $< 5\%$↑ | $< 10\%$↑ | $< 15\%$↑ | Abs Rel↓ | Sq Rel↓ | RMSE↓ | RMSE$_{log}$↓ |
|---|---|---|---|---|---|---|---|---|
| SMOKE | ✗ | 42.4% | 70.5% | 83.3% | 10.3% | 1.98 | 10.6 | 0.145 |
| RTM3D | ✗ | 42.7% | 71.1% | **83.9%** | 9.8% | 1.75 | 9.9 | 0.135 |
| R4D (**ours**) | ✓ | **46.3%** | **72.5%** | **83.9%** | **7.5%** | **0.68** | **6.8** | **0.112** |

Table 9: Comparison with monocular 3D object detection methods on pseudo long-range KITTI dataset. R4D outperforms SMOKE (Liu et al., 2020), and RTM3D (Li et al., 2020).

bels. This newly introduce Waymo Open Dataset - Long-Range Labels is a large-scale dataset with annotations for *real* long-range objects, which aligns better with the real-world scenario.

To construct the Pseudo Long-Range nuScenes Dataset, we first export the original nuScenes dataset to the KITTI format using the nuScenes offical code[1]. Then we reuse the pipeline that builds the Pseudo Long-Range KITTI Dataset. The final Pseudo Long-Range nuScenes Dataset includes 18,926 training images with target 59,800 vehicles, and 4,017 validation images with 11,737 target vehicles.

The results are shown in Table 8. R4D outperforms all baselines with a large margin. For example, R4D, outperforms Zhu & Fang (2019) (line 3) by 4.4% on the $< 10\%$↑ metric (from 66.7% to 71.1%). By comparing line 3 to line 4, we observe the performance drop when appending the LiDAR points as an additional input channel to Zhu & Fang (2019). This observation also supports our conclusion in the main manuscript: the additional distance channel does not help improve the distance estimation performance. The observation here is even more obvious, possibly because the nuScenes dataset only provides relatively low resolution LiDAR signals (32 beams, vs 64 for the KITTI Dataset) (Caesar et al., 2019).

## E    COMPARISON WITH MONOCULAR 3D OBJECT DETECTORS

In the related work section of the main manuscript, we have discussed the relationship between our proposed task, namely Long-Range Distance Estimation, and monocular 3D object detection. Here, we conduct experiments to compare R4D to monocular 3D detection methods on the Pseudo Long-Range KITTI Dataset. We associate the detected 3D bounding boxes with 2D ground-truth boxes using a bipartite matching algorithm. The weights of the bipartite graph are the 2D IoUs of ground-truth and detected boxes. In order to compare with our method, we calculate the distance prediction metrics on matched ground-truth and detected box pairs.

As shown in Table 9, R4D outperforms state-of-the-art (SOTA) 3D object detectors on the distance prediction task. For example, R4D improves the $< 5\%$↑ metric by 3.6% (from 42.7% to 46.3%) compared with the previous SOTA monocular 3D object detector RMT3D (Li et al., 2020). This improvement is more significant than the improvement from SMOKE (Liu et al., 2020) to RTM3D (Li et al., 2020) (the $< 5\%$↑ metric only improves by 0.3%) due to the fact that R4D utilizes the short-range LiDAR signals. It is also worth noting that R4D does not utilize the techniques proposed in other monocular 3D object detection papers such as multi-task training and strong architectures. Therefore, R4D is complementary to 3D detection methods, and combining it with such methods will further improve the long-range distance estimation performance.

---

[1] https://github.com/nutonomy/nuscenes-devkit/blob/master/python-sdk/nuscenes/scripts/export_kitti.py

| Valid. Set | Tgt | Ref | Uni | G-D | DA | < 15%↑ | Abs Rel↓ | Sq Rel↓ | RMSE↓ | RMSE$_{log}$↓ |
|---|---|---|---|---|---|---|---|---|---|---|
| Day | ✓ | ✗ | ✗ | ✗ | ✗ | 72.2% | 11.2% | 4.52 | 30.3 | 0.190 |
| | ✓ | ✓ | ✗ | ✗ | ✓ | 70.2% (-2.0%) | 11.2% | 4.60 (+0.08) | 30.8 (+0.5) | 0.191 (+0.001) |
| | ✓ | ✓ | ✓ | ✗ | ✓ | 73.1% (+0.9%) | 10.3% (-0.9%) | 4.03 (-0.49) | 28.9 (-1.4) | 0.178 (-0.012) |
| | ✓ | ✓ | ✓ | ✓ | ✗ | 71.5% (-0.7%) | 10.7% (-0.5%) | 4.31 (-0.21) | 29.8 (-0.5) | 0.184 (-0.006) |
| | ✓ | ✓ | ✓ | ✓ | ✓ | 77.0% (+4.8%) | 9.6% (-1.6%) | 3.71 (-0.81) | 27.8 (-2.5) | 0.171 (-0.019) |
| Dawn & | ✓ | ✗ | ✗ | ✗ | ✗ | 63.8% | 13.6% | 6.48 | 36.4 | 0.233 |
| | ✓ | ✓ | ✗ | ✗ | ✓ | 62.5% (-1.3%) | 13.5% (-0.1%) | 6.05 (-0.43) | 34.7 (-1.7) | 0.224 (-0.009) |
| | ✓ | ✓ | ✓ | ✗ | ✓ | 64.1% (+0.3%) | 13.5% (-0.1%) | 6.38 (-0.10) | 36.1 (-0.4) | 0.231 (-0.002) |
| | ✓ | ✓ | ✓ | ✓ | ✗ | 64.3% (+0.5%) | 12.8% (-0.8%) | 5.41 (-1.07) | 33.0 (-3.4) | 0.206 (-0.027) |
| Dusk | ✓ | ✓ | ✓ | ✓ | ✓ | 70.0% (+6.2%) | 11.3% (-2.3%) | 4.19 (-2.29) | 28.5 (-7.9) | 0.177 (-0.056) |
| Night | ✓ | ✗ | ✗ | ✗ | ✗ | 36.8% | 23.3% | 12.5 | 46.8 | 0.344 |
| | ✓ | ✓ | ✗ | ✗ | ✓ | 36.2% (-0.6%) | 22.9% (-0.4%) | 11.9 (-0.6) | 45.7 (-1.1) | 0.335 (-0.009) |
| | ✓ | ✓ | ✓ | ✗ | ✓ | 36.7% (-0.1%) | 22.6% (-0.7%) | 12.0 (-0.5) | 46.3 (-0.5) | 0.340 (-0.004) |
| | ✓ | ✓ | ✓ | ✓ | ✗ | 49.1% (+12.3%) | 17.5% (-5.8%) | 8.2 (-4.3) | 38.7 (-8.1) | 0.271 (-0.073) |
| | ✓ | ✓ | ✓ | ✓ | ✓ | 49.1% (+12.3%) | 17.7% (-5.6%) | 8.2 (-4.3) | 38.4 (-8.4) | 0.267 (-0.077) |

Table 10: Robustness on out-of-distribution data. All models are trained on daytime images only, and evaluated on three test sets collected during different times of day. We evaluate each components from R4D, *i.e.*, the target embeddings (Tgt), the reference embeddings (Ref), the union embeddings (Uni), the geo-distance embeddings (G-D), and distance augmentation (DA).

## F  MORE QUALITATIVE EXAMPLES ON WAYMO OPEN DATASET - LONG-RANGE LABELS

We also provide more qualitative examples of the output of the attention module in the supplemental material. All images are from our newly introduced Waymo Open Dataset - Long-Range Labels. Similar to Figure 8 in the main manuscript, the target objects are highlighted in yellow. References with the largest attention weights are highlighted in red. Other references are colored in blue. As before, we observe that the most important references are often close to or in the same lane as the target, and share a similar pose.

## G  UNDERSTANDING DISTANCE AUGMENTATION

In this section, we illustrate that the baseline model (Zhu & Fang, 2019) trained with or without distance augmentation should achieve the same objective. Intuitively, since the distance augmentation we used has zero mean ($\mu = 0$ in the Gaussian distribution), the models predict unbiased distances because they are trained to minimize the loss for all examples.

Given the same input image, the $N$ ground-truth labels (in $N$ epochs) are different due to distance augmentation. For a fixed input, the model can only predict a single value and the optimal prediction should therefore minimize the loss over all $N$ labels. To achieve the minimal loss, the model predicts the mean value over $N$ labels (if using L2 loss function) or the median value (if using L1 loss function). When $N$ is large, this mean/median value is close to the original label. To summarize, given the same input image, if we perturb the label using Gaussian noise, the model will attempt to predict the mean/median so that the loss is minimized.

Our observation from the first two lines in Table 4 also supports our intuition: the performance of the baseline models (Zhu & Fang, 2019) is roughly the same with or without distance augmentation.

## H  ABLATION STUDIES ON OUT-OF-DISTRIBUTION DATA

In Section 5.4, we demonstrate the robustness of R4D on Out-Of-Distribution data. In this section, we study the impact of each component of our proposed R4D on the Waymo Open Dataset - Long-Range Labels. The results are shown in Table 10. The observation is similar to our observation on the standard validation set (Section 5.3): each component provides a marginal improvement to model robustness, and only when used together together, is a significant improvement observed. For example, before applying Geo-distance embedding, the improvement on Abs Rel↓ on the Night set is only 0.7%, while the final R4D model improves it by 5.6%. Another interesting observation is about

distance augmentation: the model without distance augmentation does not perform well on the Day Set, but performs as well as the model with distance augmentation on the Night Set. We hypothesize that this is because the appearance information for night scenes are not as discriminative, so both models need to rely more on geometry and pairwise information. We will further investigate this in the future.

# I ROBUSTNESS AGAINST REFERENCE BOXES PERTURBATION

The goal of this section is to demonstrate how robust R4D is with respect to the noise of the input 2D bounding boxes and the object distance provided by the 3D detector. We conduct ablation studies by perturbing the 2D reference boxes (with a maximum magnitude of 15% of the box size) or the distance of the reference objects (with a maximum magnitude of 15% of the original distance). The results are shown in Tables 11 and 12. We find that, even with significant noise (*i.e.*, 15%), the performance of our model still significantly outperforms our baseline model (Zhu & Fang, 2019). For example, in Table 12, even with 15% noise, the $< 15\%\uparrow$ metric with R4D is still 4.3% better than with the baseline method (Zhu & Fang, 2019).

| Method | Noise Level | $< 5\%\uparrow$ | $< 10\%\uparrow$ | $< 15\%\uparrow$ | Abs Rel$\downarrow$ | Sq Rel$\downarrow$ | RMSE$\downarrow$ | RMSE$_{\log}\downarrow$ |
|---|---|---|---|---|---|---|---|---|
| R4D (ours) | 0% | 38.5% | 62.3% | 77.1% | 9.8% | 3.60 | 26.7 | 0.167 |
| R4D (ours) | 15% | 36.7% | 59.8% | 74.9% | 10.2% | 3.79 | 27.4 | 0.172 |
| Zhu & Fang (2019) | N/A | 31.2% | 53.4% | 68.7% | 11.5% | 4.43 | 29.4 | 0.187 |

Table 11: Noise on reference input 2D bounding boxes. We perturb the location and the size of reference 2D boxes, and then test the performance of R4D. Although slightly observing performance drop after adding noise, R4D is still better than the baseline method (Zhu & Fang, 2019).

| Method | Noise Level | $< 5\%\uparrow$ | $< 10\%\uparrow$ | $< 15\%\uparrow$ | Abs Rel$\downarrow$ | Sq Rel$\downarrow$ | RMSE$\downarrow$ | RMSE$_{\log}\downarrow$ |
|---|---|---|---|---|---|---|---|---|
| R4D (ours) | 0% | 38.5% | 62.3% | 77.1% | 9.8% | 3.60 | 26.7 | 0.167 |
| R4D (ours) | 15% | 37.8% | 60.1% | 74.2% | 10.2% | 3.82 | 27.5 | 0.172 |
| Zhu & Fang (2019) | N/A | 31.2% | 53.4% | 68.7% | 11.5% | 4.43 | 29.4 | 0.187 |

Table 12: Noise on object distance provided by the 3D detector. We perturb distance of reference 3D boxes, and then test the performance of R4D. Although slightly observing performance drop after adding noise, R4D is still better than the baseline method (Zhu & Fang, 2019).

| Method | LiDAR | $< 5\%\uparrow$ | $< 10\%\uparrow$ | $< 15\%\uparrow$ | Abs Rel$\downarrow$ | Sq Rel$\downarrow$ | RMSE$\downarrow$ | RMSE$_{\log}\downarrow$ |
|---|---|---|---|---|---|---|---|---|
| Zhu & Fang (2019) | ✗ | 31.2% | 53.4% | 68.7% | 11.5% | 4.43 | 29.4 | 0.187 |
| Zhu & Fang (2019) | ✓ | 34.1% | 56.4% | 70.5% | 11.3% | 4.78 | 31.3 | 0.194 |
| R4D (ours) | ✗ | 35.1% | 60.1% | 76.1% | 10.4% | 3.81 | 27.2 | 0.172 |
| R4D (ours) | ✓ | 38.5% | 62.3% | 77.1% | 9.8% | 3.60 | 26.7 | 0.167 |

Table 13: Results comparison on the Waymo Open Dataset - Long-Range Labels. We compare the baseline (Zhu & Fang, 2019) and R4D with and without LiDAR signals. For R4D, when LiDAR signal is missing, we train a monocular depth estimator (Zhu & Fang, 2019) on the short range objects, and then predict the distance for short-range objects as references.

# J R4D IN A PURE IMAGE-BASED SETTING

To demonstrate the potential of the concept of using reference object for depth estimation, in this section, we conduct experiments in a purely image-based setting in which the distances of reference objects are generated using a monocular depth estimation model. Specifically, we use the monocular method proposed by (Zhu & Fang, 2019) to obtain short-range distance estimates (with an 8.14% absolute relative error for short-range objects).

As shown in Table 13, in a purely image-based setting, our model significantly outperforms the baseline models. These results confirm the effectiveness of our purposed reference-based approach.

# K    MORE RELATED WORK

## K.1    LONG-RANGE DEPTH SENSING

Stereo camera systems simulate human binocular vision and use multiple cameras to create stereo-pairs and perceive depth (Khamis et al., 2018; Poggi et al., 2019). However, stereo vision systems have a few limitations: they are difficult to calibrate, are sensitive to vibrations (Zhang et al., 2020), and require a special hardware setup. In contrast, R4D does not need special hardware and works with existing sensors present on most autonomous vehicles, such as standalone camera and LiDAR. Besides, techniques used in R4D can be readily extended to other sensors and references. Radar, on the other hand, uses radio waves to determine the range, angle, and velocity of objects. However, due to its limited angular resolution (Scheiner et al., 2020), it is difficult for Radar to accurately localize long-range objects.

## K.2    MONOCULAR 3D OBJECT DETECTION

Monocular 3D object detection aims to detect all objects in an image by predicting a 3D bounding box for each of them, *e.g.*, MonoPair (Chen et al., 2020). Besides distance (from the target object to the autonomous vehicle), the monocular 3D object detector also predicts object size. For short-range vehicles, both object distance and object size are critical for self-driving tasks, such as route planning, collision avoidance, and lane changing. However, for long-range vehicles, distance is significantly larger than object size (*e.g.*, 300 meters vs. 2 meters), and predicting object size is less critical than the distance to the object. Moreover, predicting object size increases the latency and complexity of the model. *Therefore, our task focuses on estimating the distance of long-range objects.*

