# OpenReview forum: "R4D: Utilizing Reference Objects for Long-Range Distance Estimation"
_ICLR.cc/2022/Conference — ICLR 2022 Poster_

### Official Review · Reviewer_4ncP · 2021-10-18

**Correctness:** 3
**Technical Novelty And Significance:** 2
**Empirical Novelty And Significance:** 3
**Recommendation:** 6
**Confidence:** 3

**Main Review:**

## Strengths
1. The paper is well written and one can follow smoothly.
2. The idea of utilizing the known distance of the near objects to estimate the unknown distance of the distant objects is very intuitive and reasonable.
3. The network design considers various types of embeddings to explore the features from the target object, reference object, appearance relations, and geometric relations.

## Weaknesses
1. The proposed method utilizes more inputs than the other compared methods, e.g, Zhu & Fang (2019), so the better performance of R4D is expected. Specifically, R4D needs the 2D bounding boxes as well as the distance of the reference object provided by an additional 3D object detector. Also, it is not mentioned how the 2D bounding boxes are obtained: are they from another 2D detector or just projections of the 3D bounding boxes?
2. It is unclear how robust R4D is with respect to the noise of the input 2D bounding box information and the object distance provided by the 3D detector. It would be great if the authors can evaluate this aspect.
3. It is unclear whether the training/test splits are the same for all the evaluated methods.
4. The authors did not describe very clearly how the ground-truth labels for the distant objects are created. In the paper, it reads "labelers use smooth object trajectory constraints to make sure the boxes are consistent over time and space" and it would be better if the authors can elaborate more on it.
5. The distance augmentation is quite interesting. However, it can also confuse the network since the appearances of the objects should look different when the distances of the objects change. For instance, the same car should look smaller when its distance is changed from 200m to 220m. Also, when the reference object and the target object are moved together by, e.g., 20m, they will look closer on the image, which will be reflected by the union embedding.
6. In Table 4, no evaluations are shown for the experiment "We study the effect of removing the relative distance supervision" as stated in Section 5.3.
7. Since the architecture design of Attention & MLP module expects a fixed number of target-reference pairs, how it is handled in practice when there are fewer or more reference objects?

**Summary Of The Paper:**

This paper addresses the task of Long-Range Distance Estimation. In particular, it proposes R4D, a framework to estimate the distance of long-range objects by utilizing the pair-wise relations between the reference objects (objects with known distance) and the target objects (objects of which the distance is to be estimated). In addition, the authors also present two new datasets, pseudo long-range KITTI dataset and anonymous internal long-range dataset, for this task. The evaluation results show that R4D achieves better results than the previous methods.

**Summary Of The Review:**

In general, I lean towards accepting the paper since it tackles an important but underexplored task with a reasonable design of the network. Also, the proposed method achieves better results than the previous works (not a surprise though) and the authors verify the design choices with the ablation studies.

However, since I do have some concerns about the paper, my final recommendation will be heavily dependent on the responses of the authors.

---

> ### Author Response · Authors · 2021-11-23
> **Response to Reviewer 4ncP (part 1 of 2)**
>
> We thank the reviewer for the valuable comments. We addressed all the comments and revised the paper accordingly (highlighted as blue). Please check our responses below.
>
> **Q1A**: The proposed method utilizes more inputs than the other compared methods, e.g, Zhu & Fang (2019), so the better performance of R4D is expected. Specifically, R4D needs the 2D bounding boxes as well as the distance of the reference object provided by an additional 3D object detector.
>
> **Response**: Thanks for the suggestions. We conducted a set of experiments by generating depth for reference objects with the baseline method (Zhu & Fang, 2019). The performance comparison is shown as:
>
> | Method            | LiDAR |  <5%↑ | <10%↑ | <15%↑ | Abs Rel↓ | Sq Rel↓ | RMSE↓ | RMSE_log↓ |
> |-------------------|:-----:|:-----:|:-----:|:-----:|:--------:|:-------:|:-----:|:---------:|
> | Zhu & Fang (2019) |   ×   | 31.2% | 53.4% | 68.7% |   11.5%  |   4.43  |  29.4 |   0.187   |
> | Zhu & Fang (2019) |   ✓   | 34.1% | 56.4% | 70.5% |   11.3%  |   4.78  |  31.3 |   0.194   |
> | R4D (ours)        |   ×   | 35.1% | 60.1% | 76.1% |   10.4%  |   3.81  |  27.2 |   0.172   |
> | R4D (ours)        |   ✓   | 38.5% | 62.3% | 77.1% |   9.8%   |   3.60  |  26.7 |   0.167   |
>
> Even with unreliable depth for reference objects, our model still can improve the quality of the depth for target long range objects. The comparison of methods in this table are using the same amount of supervision which confirms the effectiveness of our proposed reference method. See Section J for details.
>
> **Q1B**: Also, it is not mentioned how the 2D bounding boxes are obtained: are they from another 2D detector or just projections of the 3D bounding boxes?
>
> **Response**: Just projections of the 3D bounding boxes.
>
> **Q2**. It is unclear how robust R4D is with respect to the noise of the input 2D bounding box information and the object distance provided by the 3D detector. It would be great if the authors can evaluate this aspect.
>
> **Response**: Thanks for raising this point. Ideally, more reliable object depth and 2D bounding boxes should help the network to produce more accurate depth, and noisy boxes will lead to inferior performance. To further demonstrate the impact of the quality of the 2D boxes and object distance, we conducted ablation studies with different extents of noises to boxes and distance, respectively. The results are shown as following two tables for boxes noise and distance noise:
>
> | Method            | Box Noise Level |  <5%↑ | <10%↑ | <15%↑ | Abs Rel↓ | Sq Rel↓ | RMSE↓ | RMSE_log↓ |
> |-------------------|:-----------:|:-----:|:-----:|:-----:|:--------:|:-------:|:-----:|:---------:|
> | R4D (ours)        |      0%     | 38.5% | 62.3% | 77.1% |   9.8%   |   3.60  |  26.7 |   0.167   |
> | R4D (ours)        |     15%     | 36.7% | 59.8% | 74.9% |   10.2%  |   3.79  |  27.4 |   0.172   |
> | Zhu & Fang (2019) |     N/A     | 31.2% | 53.4% | 68.7% |   11.5%  |   4.43  |  29.4 |   0.187   |
>
> | Method            | Distance Noise Level |  <5%↑ | <10%↑ | <15%↑ | Abs Rel↓ | Sq Rel↓ | RMSE↓ | RMSE_log↓ |
> |-------------------|:-----------:|:-----:|:-----:|:-----:|:--------:|:-------:|:-----:|:---------:|
> | R4D (ours)        |      0%     | 38.5% | 62.3% | 77.1% |   9.8%   |   3.60  |  26.7 |   0.167   |
> | R4D (ours)        |     15%     | 37.8% | 60.1% | 74.2% |   10.2%  |   3.82  |  27.5 |   0.172   |
> | Zhu & Fang (2019) |     N/A     | 31.2% | 53.4% | 68.7% |   11.5%  |   4.43  |  29.4 |   0.187   |
>
>
> The performance of our model slightly drops after adding noise, but still significantly outperformed our baseline method (Zhu & Fang, 2019). We have added the Tables and more explanation in the paper. More details can be found in Section I and Table 11 and 12.
>
> **Q3**. It is unclear whether the training/test splits are the same for all the evaluated methods.
>
> **Response**: For fair comparison, the same data split is used for all the experiments on each dataset by default.
>
>
> **Q4**: The authors did not describe very clearly how the ground-truth labels for the distant objects are created. In the paper, it reads "labelers use smooth object trajectory constraints to make sure the boxes are consistent over time and space" and it would be better if the authors can elaborate more on it.
>
> **Response**: In addition to LiDAR point clouds, radar signals and camera images are shown to the labelers. Instead of labeling each sample separately, labelers work on the whole sequence. Labelers first create labels for each distant object when it is shown within LiDAR range, then extend the trajectory of the distant object with additional data from radar and camera, and ensure the labels are consistent over time and space. For objects only visible in radar and camera (ie, never shown in the LiDAR range), the labels are created on a best-effort basis.

---

> > ### Author Response · Authors · 2021-11-23
> > **Response to Reviewer 4ncP (part 2 of 2)**
> >
> > **Q5**: The distance augmentation is quite interesting. However, it can also confuse the network since the appearances of the objects should look different when the distances of the objects change. For instance, the same car should look smaller when its distance is changed from 200m to 220m. Also, when the reference object and the target object are moved together by, e.g., 20m, they will look closer on the image, which will be reflected by the union embedding.
> >
> > **Response**: This question is similar to question 1 from the reviewer JYTu. To avoid repetition, please refer to the response to that question. In a nutshell, distance augmentation confuses the network on purpose, so that the network can learn more from target-reference relationships rather than solely rely on the appearance feature. More details can be found in the Section G in the appendix.
> >
> > **Q6**. In Table 4, no evaluations are shown for the experiment "We study the effect of removing the relative distance supervision" as stated in Section 5.3.
> >
> > **Response**: Thanks for noticing this point. We further conducted experiments and compared the performance of with and without using the relative distance supervision, and updated the results in Table 4. As shown in Table 4, the performance is lower when without using the relative distance supervision.
> >
> > **Q7**. Since the architecture design of the Attention & MLP module expects a fixed number of target-reference pairs, how is it handled in practice when there are fewer or more reference objects?
> >
> > **Response**: Our proposed attention module can handle a different number of target-reference pairs. The weight for the attention and mlp module are shared for each object and can be performed for each object independently. When multiple pairs are available, this data is fed as a batch to the attention and mlp module.

---

> > ### Comment · Reviewer_4ncP · 2021-11-25
> > **Thanks for the response**
> >
> > Thanks for your response and the additional experiments.
> >
> > All of my questions have been clarified and overall I'm quite satisfied with the answers. Nevertheless, I still shared a similar concern with Reviewer JYTu since the distance augmentation seems to confuse the network. But you do have, although not very strong, a point that the augmentation could prevent the network from relying too much on the image appearance instead of leveraging the relations. Also, the experimental results did show the effectiveness of such augmentation.
> >
> > Therefore, I decided to keep my rating.

---

### Official Review · Reviewer_gPz7 · 2021-10-31

**Correctness:** 4
**Technical Novelty And Significance:** 3
**Empirical Novelty And Significance:** 3
**Recommendation:** 6
**Confidence:** 5

**Main Review:**

Pros:
1. The long-range distance estimation task is practical for industry applications. It will benefit the community definitely if the proposed datasets will be released as claimed by the authors.

2. The proposed idea of extrapolating long-range distance from cues of short-range objects is great. The motivation behind the model design is also explained.

3. Experiments are sufficient. The ablation studies on data augmentation, relative distance supervision, number of reference objects are very helpful for better understanding of the contribution of each component.

4. The experiments on OOD data is great as the proposed method is shown to have better generalization ability, which is critical for autonomous driving applications.

Cons:
1. The paper uses a LiDAR-based detector to generate 3D distance of reference objects. Although the baseline method Zhu & Fang (2019) is modified to add LiDAR depth as input for fair comparison. However, I think it's equally important to experiment with pure image-based setting, i.e., generating distance of reference objects with monocular image based method. It's unclear how the quality of reference object distance affects the performance of long-range distance estimation.

2. The metrics may need further clarification. If I understood correctly, the distance metrics are computed on the set of "correctly matched" predicted objects with ground truth objects. However, the "correct match" criteria between predicted and ground truth objects is not defined, which may affect the results of distance metrics.

3. It's unclear how each component (data augmentation, contextual modeling, etc.) affects the generalization ability of the method. It would be better if more detailed analysis are added to the OOD experiments.

**Summary Of The Paper:**

The paper proposes a method for the task of long-range object distance estimation. It proposes a framework to predict long-range distance by modeling pairwise relationship between short-range objects and long-range objects. To evaluate the method, two datasets are also introduced. Experimental results are mainly compared to the baseline Zhu & Fang (2019) and several distance metrics are evaluated.

**Summary Of The Review:**

The work addresses a practical problem in autonomous driving. The motivation and the proposed method are clearly desribed. Experimental results are pretty good on two datasets. Of course, there are also room for improvement in both the experimental setting and evaluations as described in the above reviews. Nevertheless, I lean to acceptance to the paper.

---

> ### Author Response · Authors · 2021-11-23
> **Response to Reviewer gPz7**
>
> We thank the reviewer for the valuable comments. We addressed all the comments and revised the paper accordingly (highlighted as blue). Please check our responses below.
>
> **Q1**: The paper uses a LiDAR-based detector to generate 3D distance of reference objects. Although the baseline method Zhu & Fang (2019) is modified to add LiDAR depth as input for fair comparison. However, I think it's equally important to experiment with a pure image-based setting, i.e., generating distance of reference objects with a monocular image based method.
>
> **Response**: Thanks for the suggestions. We conducted a set of experiments by generating depth for reference objects with the baseline method (Zhu & Fang, 2019). The performance comparison is shown as:
>
> | Method            | LiDAR |  <5%↑ | <10%↑ | <15%↑ | Abs Rel↓ | Sq Rel↓ | RMSE↓ | RMSE_log↓ |
> |-------------------|:-----:|:-----:|:-----:|:-----:|:--------:|:-------:|:-----:|:---------:|
> | Zhu & Fang (2019) |   ×   | 31.2% | 53.4% | 68.7% |   11.5%  |   4.43  |  29.4 |   0.187   |
> | Zhu & Fang (2019) |   ✓   | 34.1% | 56.4% | 70.5% |   11.3%  |   4.78  |  31.3 |   0.194   |
> | R4D (ours)        |   ×   | 35.1% | 60.1% | 76.1% |   10.4%  |   3.81  |  27.2 |   0.172   |
> | R4D (ours)        |   ✓   | 38.5% | 62.3% | 77.1% |   9.8%   |   3.60  |  26.7 |   0.167   |
>
> Even with unreliable depth for reference objects, our model still can improve the quality of the depth for target long range objects. The comparison of methods in this table are using the same amount of supervision which confirms the effectiveness of our proposed reference method. See Section J and Table 13 for details.
>
> **Q2A**: It's unclear how the quality of reference object distance affects the performance of long-range distance estimation.
>
> **Response**: Thanks for raising this point. Ideally, more reliable object depth and 2D bounding boxes should help the network to produce more accurate depth. We conduct experiments by perturbing the reference object distance. The performance comparison is shown as:
>
> | Method            | Noise Level |  <5%↑ | <10%↑ | <15%↑ | Abs Rel↓ | Sq Rel↓ | RMSE↓ | RMSE_log↓ |
> |-------------------|:-----------:|:-----:|:-----:|:-----:|:--------:|:-------:|:-----:|:---------:|
> | R4D (ours)        |      0%     | 38.5% | 62.3% | 77.1% |   9.8%   |   3.60  |  26.7 |   0.167   |
> | R4D (ours)        |     15%     | 37.8% | 60.1% | 74.2% |   10.2%  |   3.82  |  27.5 |   0.172   |
> | Zhu & Fang (2019) |     N/A     | 31.2% | 53.4% | 68.7% |   11.5%  |   4.43  |  29.4 |   0.187   |
>
> Even with a low quality reference object distance, our model still can improve the distance estimation accuracy compared to the baseline model (Zhu & Fang, 2019). See Section I for details.
>
> **Q2B**: The metrics may need further clarification. If I understood correctly, the distance metrics are computed on the set of "correctly matched" predicted objects with ground truth objects. However, the "correct match" criteria between predicted and ground truth objects is not defined, which may affect the results of distance metrics.
>
> **Response**: To remove the impact of the quality of the 2D bounding boxes,  Zhu & Fang (2019) built the benchmark for per-object depth estimation task by using the ground-truth 2D bounding box to purely focus on the depth prediction ability. Following this benchmark and fair comparison with the other state-of-the-art methods, the 2D ground truth bounding boxes are used in all the experiments.
>
> **Q3**: It's unclear how each component (data augmentation, contextual modeling, etc.) affects the generalization ability of the method. It would be better if more detailed analysis were added to the OOD experiments.
>
> **Response**: We conducted ablation studies to demonstrate the impact of each component on the anonymous internal long-range dataset. Specifically, we evaluated the impact of the target embeddings (Tgt), the reference embeddings (Ref), the union embeddings(Uni), the geo-distance embeddings (G-D), and distance augmentation (DA), and the results are shown as Table 10 in the main paper. The observation is similar to the observation on the standard validation set: each component incurs marginal improvement on model robustness, and only when using all of these components together, a significant improvement can be observed. More details can be found in Section H.

---

### Official Review · Reviewer_EZAP · 2021-11-01

**Correctness:** 3
**Technical Novelty And Significance:** 3
**Empirical Novelty And Significance:** 3
**Recommendation:** 8
**Confidence:** 3

**Main Review:**

1. The formulation of pairwise relationships between objects is nice. It can be really hard to use a global feature representation to model the interaction between different objects/agents within the 3D scene around the self-driving vehicles. Using such a consensus-based formulation, it is easier for the framework to figure out what can be better estimated.
2. One of the weaknesses is the dependencies on accurate active sensors such as lidars and radars. This to some extent limits the application scenario for the proposed method. It might be hard to deploy the proposed method to many vehicles with limited low-level ADAS capabilities.
3. The statement of lidar sensor limitations might be inaccurate with respect to current industrial applications. Although open-source academic datasets can only provide lidar measurements for up to 80 meters, in practical industrial applications much better lidar sensors are available. Accurate measurements can be provided for at least 200 meters, even 300 meters for certain advanced models. This actually raises an interesting question as well. If lidar can generate point clouds covering a 200-meter radius, with sparser coverage at far away distances, how does this camera-based approach help in these middle-range scenarios (like 150-200 m)? For instance, we get few points covering the objects, which can lead to inaccurate depth estimation. Can the proposed method increase the estimation accuracy and robustness?

**Summary Of The Paper:**

This paper proposes to utilize camera images to further increase the perception capabilities for self-driving vehicles, especially in long-range settings. The paper builds on top of the assumption that there exist objects with accurate distance information. Then, pairwise relationships can be established between such objects and the anchoring objects. By formulating such relationships as a graph, distances can be better estimated for long-range objects.

**Summary Of The Review:**

Overall, this paper proposed an interesting solution to a very hard visual perception problem. By modeling the pairwise relationship between targets and reference objects, the proposed framework can achieve good performance for the problem of interest. The paper is well written, experimental results are solid.

---

> ### Author Response · Authors · 2021-11-23
> **Response to Reviewer EZAP**
>
> We thank the reviewer for the valuable comments. We addressed all the comments and revised the paper accordingly (highlighted as blue). Please check our responses below.
>
> **Q1**: The formulation of pairwise relationships between objects is nice. It can be really hard to use a global feature representation to model the interaction between different objects/agents within the 3D scene around the self-driving vehicles. Using such a consensus-based formulation, it is easier for the framework to figure out what can be better estimated.
>
> **Response**: Thanks for the positive feedback about our method. We demonstrated the effectiveness of using other objects as reference points for the depth estimation task on multiple datasets. We are hoping that this concept can also  be applied to other tasks in the future.
>
> **Q2**: One of the weaknesses is the dependencies on accurate active sensors such as lidars and radars. This to some extent limits the application scenario for the proposed method. It might be hard to deploy the proposed method to many vehicles with limited low-level ADAS capabilities. The statement of lidar sensor limitations might be inaccurate with respect to current industrial applications.
>
> **Response**: Thanks for your good suggestion! As we mentioned in the conclusion section, R4D is not specifically designed for LiDAR or monocular images and is readily extended to other sensors and references, such as features from high-definition maps. Applying R4D to low-level ADAS is a direction that is worth exploring as a future work.
>
> **Q3**: Although open-source academic datasets can only provide lidar measurements for up to 80 meters, in practical industrial applications much better lidar sensors are available. Accurate measurements can be provided for at least 200 meters, even 300 meters for certain advanced models. This actually raises an interesting question as well. If lidar can generate point clouds covering a 200-meter radius, with sparser coverage at far away distances, how does this camera-based approach help in these middle-range scenarios (like 150-200 m)? For instance, we get few points covering the objects, which can lead to inaccurate depth estimation. Can the proposed method increase the estimation accuracy and robustness?
>
> **Response**: This is a great point. Our proposed method can be easily generalized to these scenarios and should be able to predict more accurate depth. In the graph of our current model, the appearance and depth information are available for short range reference objects, while only the appearance information is available for target objects. In the middle-range  scenario, we can easily modify the network to also take either features extracted from the point cloud in the vicinity of the target object or the predicted depth of the target object as input. With accurate position signals from those points along with the reference objects in the graph, our method should be able to predict better depth.
>
> In our current model, we mainly use objects as the reference point and the network predicts the relationship between the target object and reference object. To apply this method to a more general setting, we can even use sparse lidar points as reference points to make the predictions. Specifically, we can select some points as landmarks and predict the relative distance of the target object to each point. In this way, our model can be easily extended to more general settings. We will study this method in the future.

---

### Official Review · Reviewer_JYTu · 2021-11-01

**Correctness:** 3
**Technical Novelty And Significance:** 2
**Empirical Novelty And Significance:** 2
**Recommendation:** 5
**Confidence:** 3

**Main Review:**

Advantages:

The paper is well written and the problem of long distance object prediction is well motivated.

The evaluation proves that using R4D shows improvement over the previous dept prediction methods.

Results on out-of-distribution data shows that the method generalizes well with different settings shows the robustness of the method.

Weaknesses:

I think there is a fundamental issue with the data augmentation. with the addition of distance to objects the relative distance between the objects remains the same but the image should also scale with the addition of such distances. how is this issue handled during the training.

How are the small objects detected?, current vision algorithms cannot detect such small objects. since the network is very dependent on such detections it would be interesting to know this problem is tackled.

what happens when there are multiple objects occluding the other object how are such scenarios tackled and how does the accuracy improve?

**Summary Of The Paper:**

The paper proposed predicting the Long-Range Distance Estimation of objects by exploiting the relative distance between objects. It demonstrated results on two novel datasets to point out the advantage of the algorithm.



**Summary Of The Review:**

The paper tackles an interesting problem of long distance estimation by introducing a novel dataset and an attention based learning framework. The authors need to clarify some my concerns before acceptance.

---

> ### Author Response · Authors · 2021-11-23
> **Response to Reviewer JYTu (part 1 of 2)**
>
> We thank the reviewer for the valuable comments. We addressed all the comments and revised the paper accordingly (highlighted as blue). Please check our responses below.
>
> **Q1**: I think there is a fundamental issue with the data augmentation. with the addition of distance to objects the relative distance between the objects remains the same but the image should also scale with the addition of such distances. How is this issue handled during the training?
>
> **Response**:
> To clarify, our distance augmentation keeps the images not changed, but the “addition value” is sampled from a Gaussian distribution, which can be positive or negative. We would like to explain the *motivation* and the *correctness* of our proposed distance augmentation.
>
> *Motivation*: we propose the distance augmentation to show a simple yet effective method to address the issue of overly relying on the appearance of objects during prediction. Here, we on purpose try to change labels inconsistent to the appearance of the target objects so that the model does not overfit to the appearance of the target object and use this appearance as a shortcut. We believe there will be other ways to address this shortcut-learning [a] problem, but if scaling the image accordingly (as suggested by the reviewer), the short-cut issue is actually not addressed.
>
> *Correctness*: Distance augmentation can be interpreted as converting ground-truth label to distribution, and this technique is also used to compute uncertainty loss [b]. We add Section G: *Understanding Distance Augmentation* in the appendix to discuss distance augmentation in detail. Specifically, we show that using or not using distance augmentation does not significantly influence the baseline models (Zhu & Fang, 2019), i.e., both ways share the same objective. Besides, our experimental results in Table 4 (line 1-2) also support our explanation – the performance of the distance estimation models does not significantly change with or without distance augmentation.
>
> To summarize, distance augmentation is a simple yet effective method to achieve our goal of guiding models that focus more on relative relationships. Advanced methods can be explored in the future.
>
> [a] Geirhos, R., Jacobsen, J.H., Michaelis, C., Zemel, R., Brendel, W., Bethge, M. and Wichmann, F.A., 2020. Shortcut learning in deep neural networks. Nature Machine Intelligence, 2(11), pp.665-673.
>
> [a] Meyer, G.P. and Thakurdesai, N., 2020. Learning an uncertainty-aware object detector for autonomous driving. In 2020 IEEE/RSJ International Conference on Intelligent Robots and Systems (IROS) (pp. 10521-10527). IEEE.
>
>
> **Q2**: How are the small objects detected? Current vision algorithms cannot detect such small objects. Since the network is very dependent on such detections it would be interesting to know if this problem is tackled.
>
> **Response**: The current 2D state-of-the-art image detectors are able to detect long-range objects. Assuming the width of a sedan is 190cm, in the Waymo Open Dataset, a 200m distance would translate to 34 pixels in width, which is large enough for general 2D object detectors. For example, even though not specifically designed for detecting small objects, YOLOv4 [a] achieves 26.7 mAP on objects smaller than 32×32 pixels on the COCO dataset. In a more general setting, higher resolution images or detection models designed for detecting small objects can be used for long-range object detection. To seek a fair comparison with Zhu & Fang (2019) , we use the same strategy to get the input bounding boxes. Moreover, many new methods which specifically focus on detecting tiny objects [b, c] have been introduced. Such methods can be employed to generate high quality boxes for tiny objects.
>
> [a] Bochkovskiy, A., Wang, C.Y. and Liao, H.Y.M., 2020. Yolov4: Optimal speed and accuracy of object detection. arXiv preprint arXiv:2004.10934.
>
> [b] Hu, P. and Ramanan, D., 2017. Finding tiny faces. In Proceedings of the IEEE conference on computer vision and pattern recognition (pp. 951-959).
>
> [c] Yu, X., Gong, Y., Jiang, N., Ye, Q. and Han, Z., 2020. Scale match for tiny person detection. In Proceedings of the IEEE/CVF Winter Conference on Applications of Computer Vision (pp. 1257-1265).

---

> > ### Author Response · Authors · 2021-11-23
> > **Response to Reviewer JYTu (part 2 of 2)**
> >
> > **Q3**: What happens when there are multiple objects occluding the other object, how are such scenarios tackled and how does the accuracy improve?
> >
> > **Response**: In this paper, we mainly focus on proposing a solution to the long-range object distance estimation. Occlusion is another critical issue for autonomous driving but not in the main scope of this paper. We conducting experiments by occluding part of the target vehicle with a black patch and the comparison with the model without occlusion is shown as:
> >
> > | Method            | Occlusion |  <5%↑ | <10%↑ | <15%↑ | Abs Rel↓ | Sq Rel↓ | RMSE↓ | RMSE_log↓ |
> > |-------------------|:---------:|:-----:|:-----:|:-----:|:--------:|:-------:|:-----:|:---------:|
> > | Zhu & Fang (2019) |     ×     | 31.2% | 53.4% | 68.7% |   11.5%  |   4.43  |  29.4 |   0.187   |
> > | Zhu & Fang (2019) |     ✓     | 26.2% | 44.8% | 59.6% |   13.8%  |   5.71  |  33.0 |   0.212   |
> > | R4D (ours)        |     ×     | 38.5% | 62.3% | 77.1% |   9.8%   |   3.60  |  26.7 |   0.167   |
> > | R4D (ours)        |     ✓     | 29.3% | 51.6% | 67.5% |   11.9%  |   4.44  |  29.0 |   0.184   |
> >
> > For both the baseline model (proposed by Zhu & Fang (2019)) and our method, the performance dropped when part of the object is occluded. This meets our expectations since the task is more challenging. Since missing occlusion labels, currently we can only construct synthetic occluded objects by adding black patches. Evaluating with real data will get more solid conclusions.

---

> > ### Comment · Reviewer_JYTu · 2021-11-25
> > **Distance augumentation clarification**
> >
> > I feel my query regarding the distance augmentation has not been addressed. I am still curious how can you augment distance to objects and still keep the image same as specified. generally any kind of label augmentation should transform image accordingly. For example, In object detection tasks augmenting the label space by flipping needs to flip the image used to train. Similarly when you add distance to the current label space you need to move either the objects in the scene to the specified augmented distance. Passing the same image by adding random distance augmentation might confuse the network. Although you show that augmentation doesn't effect the loss function in Section G, This is only valid if the input image has been repurposed to match the augmented ground-Truth distance. I still feel training the network with different distances on the same image is counter intuitive. Can you clarify whether the paper addresses this distance augmentation issues.

---

> > > ### Author Response · Authors · 2021-11-26
> > > **Your understanding is correct, but the goal of our distance augmentation is different from traditional data augmentations.**
> > >
> > > Thank you for your quick followup on this issue!
> > >
> > > We would like to first clarify that the goal of our distance augmentation is different from traditional data augmentations. Traditional data augmentation (such as flipping) is to increase the amount of data. In contrast, the goal of our distance augmentation is to address the shortcut learning issue [a].  Similar techniques have been successfully used to discourage shortcut learning in other deep learning tasks [b]. For more details, please see Section 5.1. We will make this point more clear in the next version.
> > >
> > > Then, your understanding is correct: distance augmentation will confuse the network — more specifically, during training, directly learning the absolute distance of the target object will be more difficult.
> > > This is exactly what our R4D model needed: when directly learning the absolute distance using cues from the target objects becomes difficult, R4D will learn more from the target-reference relationships since predicting the relative distances does not become difficult (we keep the relative distance not changed).
> > >
> > > To summarize, distance augmentation is designed specifically to make the target object cues (eg, the appearance of the target objects) not exactly match the absolute depth, so that relative cues are paid more attention to. Feel free to let us know if you have any questions about this.
> > >
> > > **Other minor clarifications**:
> > > “Although you show that augmentation doesn't effect the loss function in Section G, This is only valid if the input image has been repurposed to match the augmented ground-Truth distance.”
> > >
> > > **Response**: in Section G, we want to explain that the performance of the baseline model with and without distance augmentation are comparable. Even though the model will confuse due to the diverse labels for the same image (thus may lead to slower convergence speed), the baseline model can still obtain comparable performance (See line 1-2 in Table 4).
> > >
> > > All analyses in Section G are under the setting that images are not changed. We feed one input image and its $N$ distance labels ($d_1$, $d_2$, …, $d_N$) to the model as the input and the supervisions. Here $N$ is the number of training epochs, and the $N$ distance labels are generated by distance augmentation.
> > >
> > > To minimize the overall loss, the model will predict the mean (or median) distance over those $N$ distance labels, ie,
> > > $d_p = 1/N (d_1 + d_2 + … + d_N).$
> > > According to distance augmentation, $d_i (i \in [1, N])$ is sampled from Gausian distribution
> > > $\mathcal{N}(d, \sigma)$,
> > > where $d$ is the original ground-truth label. When $N$ is large, the predicted label $d_p$ will be close to the original label $d$.
> > >
> > > [a] Geirhos, R., Jacobsen, J.H., Michaelis, C., Zemel, R., Brendel, W., Bethge, M. and Wichmann, F.A., 2020. Shortcut learning in deep neural networks. Nature Machine Intelligence, 2(11), pp.665-673.
> > >
> > > [b] Li, Y., Yu, Q., Tan, M., Mei, J., Tang, P., Shen, W. and Yuille, A., 2020, September. Shape-Texture Debiased Neural Network Training. In International Conference on Learning Representations   (ICLR).

---

### Decision · Program_Chairs · 2022-01-20

**Decision:**

Accept (Poster)

**Comment:**

This paper focuses on using reference objects for long distance estimation by introducing a novel dataset and an attention based learning framework. While the presentation flows well and the methodology is practically useful, it is only marginally significant and novel. Some of the practical data augmentation aspect raise some question on whether the process wouldn't confuse the network -- the authors provide empirical evidence of the contrary in their response, although I find the principled argument to be somewhat lacking.